# AUDIT: Audio Editing by Following Instructions with Latent Diffusion Models

**Yuancheng Wang**[12][*]**, Zeqian Ju**[1]**, Xu Tan**[1]**, Lei He**[1]**, Zhizheng Wu**[2]**, Jiang Bian**[1]**, Sheng Zhao**[1]

[1]Microsoft, [2]The Chinese University of Hong Kong, Shenzhen
[1]{v-yuancwang,v-zeqianju,xuta,helei,jiabia,szhao}@microsoft.com
[2]yuanchengwang@link.cuhk.edu.cn,wuzhizheng@cuhk.edu.cn

## Abstract

Audio editing is applicable for various purposes, such as adding background sound effects, replacing a musical instrument, and repairing damaged audio. Recently, some diffusion-based methods achieved zero-shot audio editing by using a diffusion and denoising process conditioned on the text description of the output audio. However, these methods still have some problems: 1) they have not been trained on editing tasks and cannot ensure good editing effects; 2) they can erroneously modify audio segments that do not require editing; 3) they need a complete description of the output audio, which is not always available or necessary in practical scenarios. In this work, we propose **AUDIT**, an instruction-guided audio editing model based on latent diffusion models. Specifically, AUDIT has three main design features: 1) we construct triplet training data (instruction, input audio, output audio) for different audio editing tasks and train a diffusion model using instruction and input (to be edited) audio as conditions and generating output (edited) audio; 2) it can automatically learn to only modify segments that need to be edited by comparing the difference between the input and output audio; 3) it only needs edit instructions instead of full target audio descriptions as text input. AUDIT achieves state-of-the-art results in both objective and subjective metrics for several audio editing tasks (e.g., adding, dropping, replacement, inpainting, super-resolution). Demo samples are available at `https://audit-demopage.github.io/`.

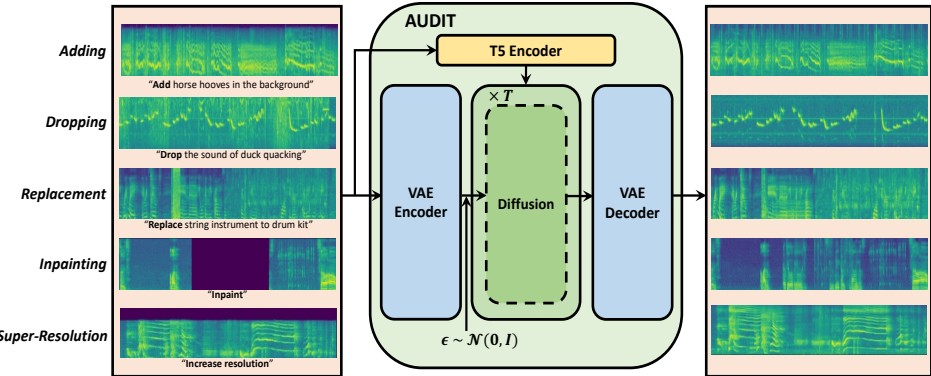

Figure 1: AUDIT consists of a VAE, a T5 text encoder, and a diffusion network, and accepts the mel-spectrogram of the input audio and the edit instructions as conditional inputs and generates the edited audio as output.

---

[*]Work done as an intern at Microsoft.

37th Conference on Neural Information Processing Systems (NeurIPS 2023).

# 1 Introduction

Audio editing is a crucial step in producing high-quality audio or video content, which involves tasks like adding background sound effects, replacing background music, repairing incomplete audio, and enhancing low-sampled audio. Typically, these tasks require professional software and manual operations. However, if audio editing tools can be empowered to follow human instructions, it could significantly reduce the need for manual intervention and benefit content creators. The utilization of natural language instructions such as "add jazz music in the background" could enable more adaptable editing in line with human expectations. In this work, we concentrate on designing audio editing models that can follow human instructions.

Previous works on audio editing usually leverage traditional signal processing technologies. Later, GAN-based methods [33, 10, 1] have achieved success in some audio editing tasks such as audio inpainting and super-resolution. However, these methods are usually designed for a specific editing task. Recently, the most advanced deep generative models, diffusion models, have achieved great success in image editing. Some methods [18, 27] have attempted to apply diffusion models to the field of audio editing. These diffusion-based audio editing methods are primarily based on a pre-trained text-to-audio generation model. However, these methods still suffer from several issues: 1) they are mainly based on a pre-trained model to noise and denoise an input audio conditioned on a target text description, and in most cases, they offer no guarantee to achieve good editing effect since they are not trained directly on audio editing tasks; 2) they can lead to erroneous modifications on some audio segments that do not need to be edited, as it is difficult to restrict the editing area; 3) they require a complete description of the output audio, which is not always available or necessary in real-world editing scenarios. Instead, it is desirable to provide natural language editing instructions such as "add a man whistling in the background" or "drop the sound of the guitar" for audio editing.

To solve the above issues, we propose AUDIT, to the best of our knowledge, the first audio editing model based on human instructions. As shown in Figure 1, AUDIT is a latent diffusion model which takes the audio to be edited and the editing instruction as conditions and generates the edited audio as output. The core designs of AUDIT can be summarized in three points: 1) we generate triplet data (instruction, input audio, output audio) for different audio editing tasks, and train an audio editing diffusion model in a supervised manner; 2) we directly use input audio as conditional input to train the model, forcing it to automatically learn to ensure that the audio segments that do not need to be edited remain consistent before and after editing; 3) our method uses editing instructions directly as text guidance, without the need for a complete description of the output audio, making it more flexible and suitable for real-world scenarios. Our editing model achieves state-of-the-art results in objective and subjective metrics.

Our contributions can be summarized as follows:

- We propose AUDIT, which demonstrates the first attempt to train a latent diffusion model conditioned on human text instructions for audio editing.
- To train AUDIT in a supervised manner, we design a novel data construction framework for each editing task; AUDIT can maximize the preservation of audio segments that do not need to be edited; AUDIT only requires simple instructions as text guidance, without the need for a complete description of the editing target.
- AUDIT achieves state-of-the-art results in both objective and subjective metrics for several audio editing tasks.

# 2 Related Works

## 2.1 Audio/Speech Editing

Previous work related to audio editing has primarily focused on human speech or music. [36, 5, 33, 55] explored the task of speech/music inpainting based on the corresponding text or music score. Another category of editing work is focused on style transfer for speech or music. Speech voice conversion [2, 35, 44, 43] and singing voice conversion [7, 28] aim to modify the timbre without changing the speech or singing content. Music style transfer [51, 9, 30, 40] aims to change the instrument being played without modifying the music score. In addition, some methods using GANs [33, 10, 1] have achieved success in some specific audio editing tasks, such as audio inpainting and audio

super-resolution. However, there is still a lack of research on general audio editing following human instructions.

## 2.2 Diffusion-based Editing

Diffusion-based editing tasks have received similar attention as diffusion-based generation tasks. We summarize the common diffusion-based editing work into two types.

**Zero-Shot Editing** Some methods [34, 8, 4, 31, 14] use diffusion-based generate models to achieve zero-shot editing. SDEdit [34] uses a pre-trained diffusion model to add noise to an input image and then denoise the image with a new target text. These methods have difficulty in editing specific regions and have poor controllability. To edit specific regions, [4] uses an additional loss gradient to guide the sampling process, and [4, 31] replace unmodified parts with the noisy input image in each sampling step, equivalent to only generating in the masked part. However, these methods are only for inpainting tasks and also need a complete description of the editing target.

**Supervised Training** Another type of methods [47, 6, 54, 37] handle editing tasks in a supervised manner. [47] train an image-to-image diffusion model which takes the image to be edited as a condition. [47] uses paired image data generated by [14] to train a text-based image editing model. ControlNet [54] and T2I-Adapter [37] add an adapter model to a frozen pre-trained generation diffusion model and train the adapter model to achieve image editing. Our method follows the supervised training paradigm, using the generated triplet data (instruction, input audio, output audio) to train an audio editing model conditioned on the instruction and the audio to be edited, and can handle different audio editing tasks by following human instructions.

## 2.3 Text-Guided Audio Generation

Currently, most diffusion-based audio editing methods [18, 27] are based on text-to-audio diffusion generative models. Text-to-audio generation aims to synthesize general audio that matches the text description. DiffSound [52] is the first attempt to build a text-to-audio system based on a discrete diffusion model [13]. AudioGen [26] is another text-to-audio generation model based on a Transformer-based decoder that uses an autoregressive structure. AudioGen directly predicts discrete tokens obtained by compressing from the waveform [53]. Recently, Make-an-Audio [18] and AudioLDM [27] have attempted to build text-guided audio generation systems in continuous space, which use an autoencoder to convert mel-spectrograms into latent representation, build a diffusion model that works on the latent space, and reconstruct the predicted results of the latent diffusion model as mel-spectrograms using the autoencoder. Furthermore, MusicLM [3] and Noise2Music [17] generate music that matches the semantic information in the input text. In our work, we focus on editing the existing audio with human instructions.

# 3 Method

In this section, we present our method for implementing audio editing based on human instructions. We provide an overview of our system architecture and training objectives in Section 3.1 and show the details about how we generate triplet data (instruction, input audio, output audio) for each editing task in Section 3.2. Lastly, we discuss the advantages of our method in Section 3.3.

## 3.1 System Overview

Our system consists of an autoencoder that projects the input mel-spectrograms to an efficient, low-dimensional latent space and reconstructs it back to the mel-spectrograms, a text encoder that encodes the input text instructions, a diffusion network for editing in the latent space, and a vocoder for reconstructing waveforms. The overview of the system is shown in Figure 2.

**Autoencoder** The autoencoder contains an encoder $E$ and a decoder $G$. The encoder $E$ transforms the mel-spectrogram $x$ into a latent representation $z$, and the decoder $G$ reconstructs $\hat{x}$ from the latent space. In our work, we employ a variational autoencoder (VAE) [23] model as the autoencoder. We train our autoencoder with the following loss: 1) The $\mathcal{L}_1$ and $\mathcal{L}_2$ reconstruction loss. 2) The

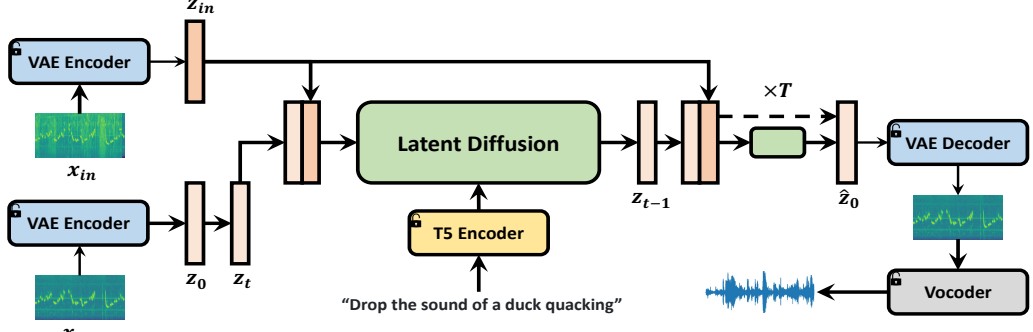

Figure 2: A high-level overview of our system.

Kullback-Leibler loss $\mathcal{L}_{KL}$ to regularize the latent representation $z$. We only assign a small weight $\lambda_{KL}$ to the KL loss following [45]; 3) The GAN loss $\mathcal{L}_{GAN}$. We employ a patch-based discriminator $D$ [19, 45] to distinguish between real and reconstructed mel-spectrograms. The total training loss of the autoencoder can be expressed as $\mathcal{L}_{VAE} = \lambda_1\mathcal{L}_1 + \lambda_2\mathcal{L}_2 + \lambda_{KL}\mathcal{L}_{KL} + \lambda_{GAN}\mathcal{L}_{GAN}$. $\lambda_1$, $\lambda_2$, $\lambda_{KL}$, and $\lambda_{GAN}$ are the weights of $\mathcal{L}_1$, $\mathcal{L}_2$, $\mathcal{L}_{KL}$, and $\mathcal{L}_{GAN}$ respectively.

**Text Encoder**   We use a pre-trained language model T5 [21] as our text encoder, which is used to convert text input into embeddings that contain rich semantic information. The parameters of the text encoder are frozen in the training stage.

**Latent Diffusion Model**   Our text-guided audio editing model can be seen as a conditional latent diffusion model that takes the latent representation $z_{in}$ of the input (to be edited) mel-spectrogram $x_{in}$ and text embeddings as conditions. The model aims to learn $p(z_{out}|z_{in}, c_{text})$, where $z_{out}$ is the latent representation of the output (edited) mel-spectrogram $x_{out}$, and $c_{text}$ is the embedding of the editing instruction. Given the latent representation $z_{in}$ and the text instruction, we randomly select a time step $t$ and a noise $\epsilon$ to generate a noisy version $z_t$ of the latent representation $z_{out}$ by using a noise schedule and then use the diffusion network $\epsilon_{\theta}(z_t, t, z_{in}, c_{text})$ to predict the sampled noise. The training loss is $\mathcal{L}_{LDM} = \mathbf{E}_{(z_{in}, z_{out}, text)}\mathbf{E}_{\epsilon \sim \mathcal{N}(0, I)}\mathbf{E}_t||\epsilon_{\theta}(z_t, t, z_{in}, c_{text}) - \epsilon||_2$.

Similarly to the original standard diffusion model [15, 49], we use a U-Net [46] with the cross-attention mechanism [50] as the diffusion network. Our editing model takes $z_{in}$ as a condition by directly concatenating $z_t$ and $z_{in}$ at the channel level. Therefore, the input channel number of the first layer of the U-Net is twice the output channel number of the last layer.

**Vocoder**   A vocoder model is required to convert the output mel-spectrogram into audio. In this work, we use HiFi-GAN [24] as the vocoder, which is one of the most widely used vocoders in the field of speech synthesis. HiFi-GAN uses multiple small sub-discriminators to handle different cycle patterns. Compared with some autoregressive [38, 20] or flow-based [42] vocoders, it takes into account both generation quality and inference efficiency.

### 3.2   Generating Triplet Training Data for Each Editing Task

We use generated triplet data (instruction, input audio, output audio) to train our text-guided audio editing model. In this work, we focus on five different editing tasks, including adding, dropping, replacement, inpainting, and super-resolution. Note that we train all the editing tasks in a single editing model. Figure 3 provides an overview of how our data generation workflow for different editing tasks works.

**Adding**   We randomly select two audio clips $A$ and $B$ from the text-audio datasets, then combine $A$ and $B$ to get a new audio clip $C$. We use $A$ as the input audio and $C$ as the output audio, and we fill in the caption (or label) of $B$ into the instruction template to get the instruction. For example, as shown in Figure 3, the instruction template is *"Add {} in the background"* and the caption of $B$

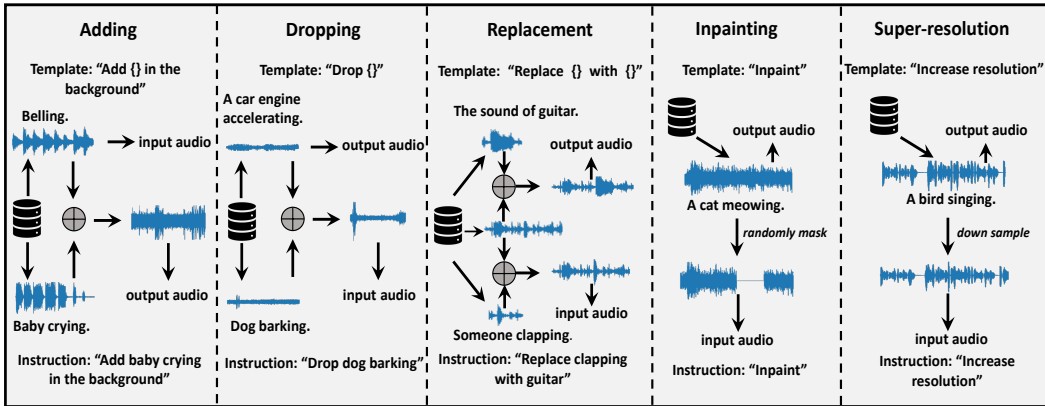

Figure 3: Examples about how to generate triplet training data for different audio editing tasks (adding, dropping, replacement, inpainting, and super-resolution).

is *"Baby crying"*, so the instruction is *"Add baby crying in the background"*. Note that we design multiple templates for different editing tasks.

**Dropping**  We randomly select two audio clips $A$ and $B$ from the text-audio datasets, then combine $A$ and $B$ to get a new audio clip $C$. We use $C$ as the input audio and $A$ as the output audio, and we fill in the caption (or label) of $B$ into the instruction template to get the instruction. For example, the instruction template is *"Drop {}"* and the caption of $B$ is *"Dog barking"*, so the instruction is *"Drop Dog barking"*.

**Replacement**  For the replacement task, we select three audio clips $A$, $B$, and $C$ from the datasets, then insert $B$ and $C$ separately into $A$ (in roughly the same area) to obtain two new audio clips $D$ and $E$. We use $D$ as the input audio and $E$ as the output audio, and we fill in the captions (or labels) of $B$ and $C$ into the instruction template to get the instruction. For example, the instruction template is *"Replace {} with {}"*, the caption of $B$ is *"Someone clapping"*, and the caption of $C$ is *"The sound of guitar"*, so the instruction is *"Replace clapping with guitar"*.

**Inpainting and Super-Resolution**  For the inpainting tasks, we select an audio clip as the output audio and randomly mask some parts of the audio to get a new audio clip as the input audio. We can use instructions like *"Inpaint"* or *"Inpaint the audio"* directly or use the instructions like *"Inpaint: a cat meowing"*. For the super-resolution tasks, we use down-sampled audio as the input audio. The instructions are as *"Increase resolution"* or *"Increase resolution: a bird singing"*.

**Extending Instructions with ChatGPT**  We design multiple instruction templates for each editing task. In order to further expand the diversity of editing instructions, we use ChatGPT [39] to generate more editing instruction templates for us. See more details in Appendix E.

### 3.3  Advantages of AUDIT

We analyze the advantages of our methods over previous works on three key points. 1) We generated triplet data (instruction, input data, output data) to train a text-guided audio editing model instead of performing zero-shot editing, which can ensure good edit quality. 2) We directly use the input audio as a condition for supervised training of our diffusion model, allowing the model to automatically learn to preserve the parts that do not need to be edited before and after editing. Specifically, we concatenate the latent representations $z_{in}$ of the input audio and $z_t$ on the channel dimension and input them into the latent diffusion model, so that the model can "see" $z_{in}$ (rather than its noisy version) during both training and inference. 3) Instead of using a full description of the output audio, we use human instructions as the text input, which is more in line with real-world applications.

# 4 Experimental Settings

## 4.1 Dataset

The datasets used in our work consist of AudioCaps [22], AudioSet [12], FSD50K [11], and ESC50 [41]. AudioSet is the largest audio-label pairs dataset; however, each audio clip in AudioSet only has a corresponding label and no description. Because of the presence of a large number of human speech and some audio clips that are only weakly related to their labels in AudioSet, we use a subset of AudioSet (AudioSet96) that includes approximately 149K pairs with 96 classes. AudioCaps is a dataset of around 44K audio-caption pairs, where each audio clip corresponds to a caption with rich semantic information. The length of each audio clip in AudioSet and AudioCaps is around 10 seconds. FSD50K includes around 40K audio clips and 200 audio-label pairs of variable lengths ranging from 0.3 to 30 seconds. We split FSD50K training set into two datasets, FSD50K-L (19K) and FSD50K-S (22K), comprising audio clips of length less than and greater than 5 seconds, respectively. ESC50 is a smaller dataset with 50 classes, each containing four audio clips of around 5 seconds, for a total of 2000 audio clips.

We use AudioCaps, AudioSet96, FSD50K-S, and ESC50 to generate triplet training data for five audio editing tasks. We use a total of about 0.6M triplet data to train our audio editing model. More details about datasets and data processing are shown in Appendix A.

## 4.2 Baseline Systems

**Text-to-Audio Generation**    Since we compare with generative model-based audio editing baseline methods, we use AudioCaps, AudioSet96, FSD50K, and ESC50 to train a text-to-audio latent diffusion model and use the test set of AudioCaps to evaluate the text-to-audio model. To demonstrate the performance of our generative model, we compare it with some state-of-the-art text-to-audio generative models [52, 26, 27, 18].

**Adding, Dropping, and Replacement**    We use methods like SDEdit[34] as baselines. SDEdit uses a pre-trained text-to-image diffusion model to noise and then denoise an input image with the editing target description. In our work, we directly use SDEdit in the adding task, dropping task, and replacement task. We use our own trained text-to-audio latent diffusion model. We test different total denoising steps $N$ for $N = 3/4T$, $N = 1/2T$, and $N = 1/4T$, where $T$ is the total step in the forward diffusion process. The three baselines are called: 1) *SDEdit-3/4T*; 2) *SDEdit-1/2T*; 3) *SDEdit-1/4T*.

**Inpainting**    For the inpainting task, we designed four baselines derived from SDEdit but with some differences between each other. 1) *SDEdit*. We directly use SDEdit, we first add noise to the input audio, then denoise the audio with the description (caption or label) of the output audio as text guidance. 2) *SDEdit-Rough* and 3) *SDEdit-Precise*. Only a part of the input audio (the part that is masked) needs to be edited in the inpainting task, we call the part that does not need to be edited the "observable" part, and we call the masked part the "unobservable" part. In each step of the denoising process, we can replace the "observable" part with the ground truth in the latent space. The difference between SDEdit-Rough and SDEdit-Precise is that in SDEdit-Rough, the "unobservable" part is a rough region, while in SDEdit-Precise, the "unobservable" part is a precise region. 4) *SDEdit-wo-Text*. SDEdit-wo-Text is similar to SDEdit-Precise, however, it has no text input. An example of the difference of the "unobservable" part between SDEdit-Rough and SDEdit-Precise is shown in Appendix F. We also compared our methods with two task-specific methods GACELA [32] and DAS [10]. GACELA is a generative adversarial network for restoring missing musical audio data. The publicly available checkpoint was trained solely on music data; hence, we trained the model using our own dataset. DAS is another GAN-based audio editing model, we also trained it on our dataset.

**Super-Resolution**    Super-resolution can be seen as inpainting in the frequency domain. We use three baselines similar to the inpainting task: 1) *SDEdit*; 2) *SDEdit-Precise*; 3) *SDEdit-wo-Text*. We also conducted a comparison with the NU-Wave2 [51] model, which is a task-specific model designed for super-resolution. We utilized the provided open-source checkpoint for NU-Wave2; however, note that NU-Wave2 was originally trained on speech data. To adapt it to our specific task, we trained it on our own dataset.

### 4.3 Evaluation Metrics

**Objective Metrics**   We use log spectral distance (LSD), frechet distance (FD), and kullback–leibler divergence (KL) to evaluate our text-guided audio editing model and use inception score (IS), FD, and KL to evaluate our text-to-audio generation model following [52, 26, 27, 18]. LSD measures the distance between frequency spectrograms of output samples and target samples. FD measures the fidelity between generated samples and target samples. IS measures the quality and diversity of generated samples. KL measures the correlation between output samples and target samples. FD, IS, and KL are based on the state-of-the-art audio classification model PANNs [25]. We use the evaluation pipeline [2] provided by [27] for objective evaluation for fairer comparisons.

**Subjective Metrics**   We use overall quality (Quality) to measure *the sound quality and naturalness of the output audio compared to the input audio* and use relevance to the editing instruction (Relevance) to measure *how well the output audio matches the input human instruction*. Each sample will be scored from 1 to 100 based on Quality and Relevance. See more details about the evaluation in Appendix D.

### 4.4 Model Configurations

Our autoencoder compresses a mel-spectrogram of size $1 \times H \times W$ into a latent representation of size $4 \times \frac{H}{4} \times \frac{W}{4}$. Our models are trained on 8 NVIDIA V100 GPUs for 500K steps with a batch size of 2 on each device. We use the weights of our pre-trained text-to-audio model to initialize our audio editing model. Our HiFi-GAN vocoder is trained on AudioSet96 and AudioCaps datasets using 8 NVIDIA V100 GPU for 200 epochs. More details about model configurations are shown in Appendix B.

## 5   Results

### 5.1   Objective Evaluation

**Adding**   Table 1 shows the objective evaluation results of the adding task. Our method achieves the best performance, with LSD of 1.35, KL of 0.92, and FD of 21.80. Compared with the best baseline method, our method reduces FD by 6.45 and KL by 0.38.

Table 1: Objective evaluation results of the adding task.

| Model | Text | LSD($\downarrow$) | KL($\downarrow$) | FD($\downarrow$) |
|---|---|---|---|---|
| SDEdit-3/4T | caption | 1.54 | 1.68 | 28.87 |
| SDEdit-1/2T | caption | 1.43 | 1.38 | 28.75 |
| SDEdit-1/4T | caption | 1.38 | 1.30 | 28.25 |
| AUDIT | instruction | **1.35** | **0.92** | **21.80** |

**Dropping**   Table 2 shows the objective evaluation results of the dropping task. Our method achieves the best performance, with LSD of 1.37, KL of 0.95, and FD of 22.40. Compared with the best baseline method, our method reduces FD by 5.79 and KL by 0.10.

Table 2: Objective evaluation results of the dropping task.

| Model | Text | LSD($\downarrow$) | KL($\downarrow$) | FD($\downarrow$) |
|---|---|---|---|---|
| SDEdit-3/4T | caption | 1.54 | 1.14 | 29.66 |
| SDEdit-1/2T | caption | 1.43 | 1.05 | 28.19 |
| SDEdit-1/4T | caption | 1.40 | 1.30 | 31.31 |
| AUDIT | instruction | **1.37** | **0.95** | **22.40** |

---

[2]`https://github.com/haoheliu/audioldm_eval`

**Replacement**  Table 3 shows the objective evaluation results of the replacement task. Our method achieves the best performance, with LSD of 1.37, KL of 0.84, and FD of 21.65. Compared with the best baseline method, our method reduces FD by 5.07 and KL by 0.31.

Table 3: Objective evaluation results of the replacement task.

| Model | Text | LSD($\downarrow$) | KL($\downarrow$) | FD($\downarrow$) |
|---|---|---|---|---|
| SDEdit-3/4T | caption | 1.63 | 1.58 | 28.78 |
| SDEdit-1/2T | caption | 1.52 | 1.27 | 27.71 |
| SDEdit-1/4T | caption | 1.46 | 1.15 | 26.72 |
| AUDIT | instruction | **1.37** | **0.84** | **21.65** |

**Inpainting**  Table 4 shows the objective evaluation results of the inpainting task. Our method achieves the best performance, with LSD of 1.32, KL of 0.75, and FD of 18.17. We find that for baseline methods, not providing a text input (the caption of the audio) as guidance leads to a large performance drop, SDEdit-wo-Text gets the worst performance in terms of KL and FD among the baseline methods. However, AUDIT-wo-Text which only uses instructions like *"inpaint"* achieves performance close to AUDIT which uses instructions like *"inpaint + caption"*. We also discovered that AUDIT's performance on inpainting tasks can surpass that of task-specific baseline methods. One possible explanation for this is that during training, GACELA and DAS do not have access to any textual information, whereas our model requires explicit instructions to edit the audio. As a result, our model is better able to learn the semantic information of the audio, which likely contributes to its improved performance in inpainting tasks.

Table 4: Objective evaluation results of the inpainting task.

| Model | Text | LSD($\downarrow$) | KL($\downarrow$) | FD($\downarrow$) |
|---|---|---|---|---|
| SDEdit | caption | 2.91 | 1.47 | 25.42 |
| SDEdit-Rough | caption | 1.64 | 0.98 | 21.99 |
| SDEdit-Precise | caption | 1.54 | 0.94 | 21.07 |
| SDEdit-wo-Text | - | 1.55 | 1.63 | 27.63 |
| GACELA | - | 1.41 | 0.78 | 20.49 |
| DAS | - | 1.57 | 0.89 | 21.97 |
| AUDIT-wo-Text | instruction | 1.37 | 0.81 | 19.03 |
| AUDIT | instruction + caption | **1.32** | **0.75** | **18.17** |

**Super-Resolution**  Table 5 shows the objective evaluation results of the super-resolution task. Our method achieves the best performance, with LSD of 1.48, KL of 0.73, and FD of 18.14. Similar to the inpainting task, SDEdit-wo-Text gets the worst performance in terms of KL and FD among the baseline methods. Our method can achieve significantly better results than baselines using only simple instructions like *"Increase resolution"*, which shows that our method can learn sufficient semantic information from low-frequency information. The results also indicate that our AUDIT model outperforms the NU-Wave2 model trained on speech and is comparable to or even better than the NU-Wave2 model trained on our dataset.

**Text-to-Audio Generation**  We also present the comparison results of our text-to-audio latent diffusion model with other text-to-audio models in Table 6. Our model achieves the best performance, with FD of 20.19, KL of 1.32, and IS of 9.23, outperforming AudioLDM-L-Full with FD of 23.31, KL of 1.59, and IS of 8.13. This demonstrates that our generation model can serve as a strong baseline model for generation-based editing methods.

## 5.2  Subjective Evaluation

The results of the subjective evaluation are shown in Table 7. We choose the best results in the baseline to report. Our method clearly outperforms the baseline methods in both Quality and Relevance across

Table 5: Objective evaluation results of the super-resolution task.

| Model | Text | LSD(↓) | KL(↓) | FD(↓) |
|-------|------|--------|-------|-------|
| SDEdit | caption | 3.14 | 1.50 | 25.31 |
| SDEdit-Precise | caption | 1.75 | 1.17 | 27.81 |
| SDEdit-wo-Text | - | 1.78 | 1.56 | 32.66 |
| NU-Wave2 (origin) | - | 1.66 | 0.89 | 22.61 |
| NU-Wave2 | - | **1.42** | 1.78 | 19.57 |
| AUDIT-wo-Text | instruction | 1.53 | 0.92 | 21.97 |
| AUDIT | instruction + caption | 1.48 | **0.73** | **18.14** |

Table 6: The comparison between our text-to-audio generative model and other baseline models on the AudioCaps test set.

| Model | Dataset | FD(↓) | IS(↑) | KL(↓) |
|-------|---------|-------|-------|-------|
| DiffSound [52] | AudioSet+AudioCaps | 47.68 | 4.01 | 2.52 |
| AudioGen [26] | AudioSet+AudioCaps | - | - | 2.09 |
| Make-an-Audio[18] | AudioSet+AudioCaps+13 others | - | - | 2.79 |
| AudioLDM-L [27] | AudioCaps | 27.12 | 7.51 | 1.86 |
| AudioLDM-L-Full [27] | AudioSet+AudioCaps+2 others | 23.31 | 8.13 | 1.59 |
| AUDIT | AudioSet96+AudioCaps+2 others | **20.19** | **9.23** | **1.32** |

all five tasks. In the dropping task, our method achieves the highest scores with Quality of 78.1 and Relevance of 81.0. We find that compared with the baseline methods, our method improves more significantly on the adding, dropping, and replacement tasks. The possible reason is that compared with the inpainting and super-resolution tasks, which have explicit positioning of the editing area (the masked part and the high-frequency area), the adding, dropping, and replacement tasks need to first locate the region to be edited, and also need to ensure that other regions cannot be modified, which is more difficult for the baseline method. Our model has learned this ability through supervised learning.

Table 7: Subjective evaluation. For each audio editing task, we report the results of the best baseline method and the results of our method.

| Method | Task | Quality(↑) | Relevance(↑) |
|--------|------|------------|--------------|
| Baseline | Adding | 60.2 | 56.7 |
| AUDIT | | **75.5** | **77.3** |
| Baseline | Dropping | 54.4 | 48.2 |
| AUDIT | | **78.1** | **81.0** |
| Baseline | Replacement | 57.7 | 47.6 |
| AUDIT | | **72.5** | **74.5** |
| Baseline | Inpainting | 65.8 | 66.3 |
| AUDIT | | **75.2** | **78.7** |
| Baseline | Super-Resolution | 61.8 | 59.9 |
| AUDIT | | **74.6** | **76.3** |

## 5.3 Case Study

In order to more specifically show the difference between the performance of our model and the baseline methods, we also give some case studies in Appendix G.

# 6 Conclusions

In this work, we propose an audio editing model called AUDIT, which can perform different editing tasks (e.g., adding, dropping, replacement, inpainting, super-resolution) based on human text instructions. Specifically, we train a text-guided latent diffusion model using our generated triplet training data (instruction, input audio, output audio), which only requires simple human instructions as guidance without the need for the description of the output audio and performs audio editing accurately without modifying audio segments that do not need to be edited. AUDIT achieves state-of-the-art performance on both objective and subjective metrics for five different audio editing tasks. For future work, we will explore more editing audio tasks with our framework, and achieve more precise control for audio editing. We also discuss some limitations and broader impacts of our work in Appendix H.

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

## A  Datasets and Data Processing

Table 8 presents some details about the datasets we used and generated. All audio data have a sampling rate of 16KHz. For AudioSet, AudioCaps, and AudioSet96, we pad or truncate all audio to 10s. For FSD50K, we pad audio shorter than 5s to 5s, and truncate or pad audio longer than 5s to 10s. All generated audio data have a length of 10 seconds. We convert 10s of audio into mel-spectrograms with a size of $80 \times 624$, using a hop size of 256, a window size of 1024, and mel-bins of size 80, covering frequencies from 0 to 8000. The autoencoder compresses the mel-spectrograms to a latent representation of size $4 \times 10 \times 78$.

Table 8: Details about audio-text datasets we use and our generated audio editing datasets

| Dataset | Hours | Number | Text |
|---|---|---|---|
| AudioSet | 5800 | 2M | label |
| AudioSet96 | 414 | 149K | label |
| AudioCaps | 122 | 44K | caption |
| FSD50K | 108 | 51K | label |
| FSD50K-S | 31 | 22K | label |
| FSD50K-L | 53 | 19K | label |
| ESC50 | 3 | 2K | label |

| Task | Datasets | Number | Text |
|---|---|---|---|
| Generation | AudioCaps, AudioSet96, FSD50K, ESC50 | 243K | label or caption |
| Adding | AudioCaps, AudioSet96, FSD50K-S,ESC50 | 71K | Instruction |
| Dropping | AudioCaps, AudioSet96, FSD50K-S,ESC50 | 71K | Instruction |
| Replacement | AudioSet96, FSD50K-S, ESC50 | 50K | Instruction |
| Inpainting | AudioSet96, AudioCaps | 193K | Instruction |
| Super-resolution | AudioSet96, AudioCaps | 193K | Instruction |

## B  Model Details

Table 9 shows more details about our audio editing and audio generative models. We train our autoencoder model with a batch size of 32 (8 per device) on 8 NVIDIA V100 GPUs for a total of 50000 steps with a learning rate of $7.5e - 5$. For both audio editing and U-Net audio generative diffusion, we train with a batch size of 8 on 8 NVIDIA V100 GPUs for a total of 500000 steps with a learning rate of $5e - 5$. Both the autoencoder and diffusion models use AdamW[29] as the optimizer with $(\beta_1, \beta_2) = (0.9, 0.999)$ and weight decay of $1e - 2$. We follow the official repository to train the HiFi-GAN vocoder.

## C  Classifier-free Guidance

[16] proposed using classifier-free guidance to trade off diversity and sample quality. The classifier-free guidance strategy has been widely used in conditional diffusion models. Our model has two additional conditions, $c_{text}$ and $z_{in}$. However, during training, we only consider helping the model learn the marginal distribution of $p(z_t|z_{in})$ (without explicitly learning $p(z_t|c_{text})$). Therefore, during training, we mask the text with a certain probability (replacing it with an empty text $\emptyset$) to learn the no-text condition score $\epsilon_\theta(z_t, t, z_{in})$. Then, according to Bayes' formula $p(z_t|c_{text}) \propto \frac{p(z_t|c_{text}, z_{in})}{p(z_{in}|z_t, c_{text})}$, we can derive the score relationship in Equation 1, which corresponds to the classifier-free guidance Equation 2. Here, the parameter $s \geq 1$ is the guidance coefficient used to balance the diversity and quality of the samples.

$$\nabla_{z_t} \log p(z_{in}|z_t, c_{text}) = \nabla_{z_t} \log p(z_t|c_{text}, z_{in}) - \nabla_{z_t} \log p(z_t|z_{in}) \tag{1}$$

$$\tilde{\epsilon}_\theta(z_t, t, z_{in}, c_{text}) = \epsilon_\theta(z_t, t, z_{in}, \emptyset) + s \cdot (\epsilon_\theta(z_t, t, z_{in}, c_{text}) - \epsilon_\theta(z_t, t, z_{in}, \emptyset)) \tag{2}$$

Table 9: Details about our audio editing and audio generative models

| Model | Configuration | |
|---|---|---|
| Autoencoder | Number of Parameters | 83M |
| | In/Out Channels | 1 |
| | Latent Channels | 4 |
| | Number of Down/Up Blocks | 4 |
| | Block Out Channels | (128, 256, 512, 512) |
| | Activate Function | SiLU |
| T5 Text Encoder | Number of Parameters | 109M |
| | Output Channels | 768 |
| | Max Length | 300 |
| Editing Diffusion U-Net | Number of Parameters | 859M |
| | In Channels | 8 |
| | Out Channels | 4 |
| | Number of Down/Up Blocks | 4 |
| | Block Out Channels | (320, 640, 1280, 1280) |
| | Attention Heads | 8 |
| | Cross Attention Dimension | 768 |
| | Activate Function | SiLU |
| Generative Diffusion U-Net | Number of Parameters | 859M |
| | In Channels | 4 |
| | Out Channels | 4 |
| | Number of Down/Up Blocks | 4 |
| | Block Out Channels | (320, 640, 1280, 1280) |
| | Attention Heads | 8 |
| | Cross Attention Dimension | 768 |
| | Activate Function | SiLU |
| HiFi-GAN | Sampling Rate | 16000 |
| | Number of Mels | 80 |
| | Hop Size | 256 |
| | Window Size | 1024 |

## D  Human Evaluation

For each audio editing task, our human evaluation set comprises ten samples randomly selected from the test set. Ten raters score each sample according to two metrics, Quality and Relevance, using a scale of 1 to 100.

## E  Extending Instructions with ChatGPT

Some examples of instruction templates designed by ourselves:

*"Add {} in the beginning"*

*"Add {} at the beginning"*

*"Add {} in the end"*

*"Add {} in the middle"*

*"Add {} in the background"*

*"Drop {}"*

*"Remove {}"*

*"Replace {} to {}"*

*"Replace {} with {}"*

*"Inpaint"*

*"Inpainting"*

*"Inpaint {}"*

*"Inpaint: {}"*

*"Increase resolution"*

*"Increase resolution: {}"*

*"Perform super-resolution"*

*"Perform super-resolution: {}"*

We use ChatGPT to extend editing templates, specifically, we submit the editing instruction templates we designed to ChatGPT and let it generate more instruction templates with the same semantic information. Below are some examples.

*"Mix {} into the background"*

*"Blend {} with existing audio"*

*"Incorporate {} as a new element at the end of the audio"*

*"Place {} in the foreground"*

*"Erase {} from the track"*

*"Subtract {} from the audio"*

*"Take out {} from the foreground"*

*"Exchange {} for {} in the mix"*

*"Use {} to replace {} in the audio"*

*"Interchange {} and {} in the track"*

*"Replace missing audio with synthesized sound"*

*"Fill in the gaps in track {}"*

*"Upscale audio to higher resolution"*

*"Apply super-resolution to the audio to improve clarity"*

## F    Baseline Methods

For the inpainting task, only a part of the input audio (the part that is masked) needs to be edited, we call the part that does not need to be edited the "observable" part, and we call the masked part the "unobservable" part. In each step of the denoising process, we can replace the "observable" part with the ground truth in the latent space. The difference between SDEdit-Rough and SDEdit-Precise is that in SDEdit-Rough, the "unobservable" part is a rough region, while in SDEdit-Precise, the "unobservable" part is a precise region. Figure 4 gives an example.

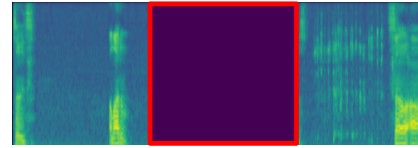
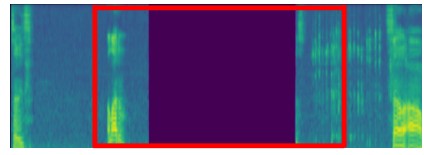

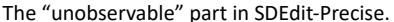

The "unobservable" part in SDEdit-Precise.          The "unobservable" part in SDEdit-Rough.

Figure 4: The difference between SDEdit-Rough and SDEdit-Precise.

## G  Case Study

We show case studies. Figure 5 shows a case for the adding task. The caption of the input audio is *"The sound of machine gun"*, and the editing target is adding a bell ringing in the beginning. AUDIT performs audio editing accurately in the correct region without modifying audio segments that do not need to be edited. Figure 6 shows a case for the dropping task. The caption of the input audio is *"A bird whistles continuously, while a duck quacking in water in the background"*, and the editing target is dropping the sound of the duck quacking. Our method successfully removes the background sound and preserves the sound of the bird whistling, but the sound of the bird whistling in the SDEdit editing result is incorrectly modified. It shows that our method can better ensure that the audio segments that do not need to be edited are not modified. Figure 7 shows a case for the inpainting task. The caption of the input audio is *"A person is typing computer"*. While both AUDIT and the baseline method generate semantically correct results, the result generated by AUDIT is more natural and contextual.

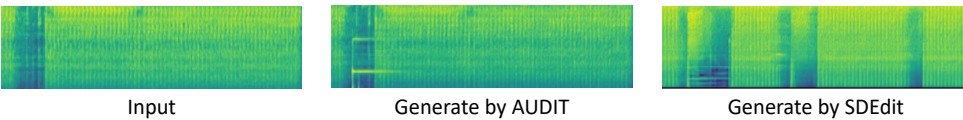

Figure 5: A case for the adding task. The caption of the input audio is *"The sound of machine gun"*, and the editing target is adding a bell ringing in the beginning.



Figure 6: A case for the dropping task. The caption of the input audio is *"A bird whistles continuously, while a duck quacking in water in the background"*, and the editing target is dropping the sound of the duck quacking.

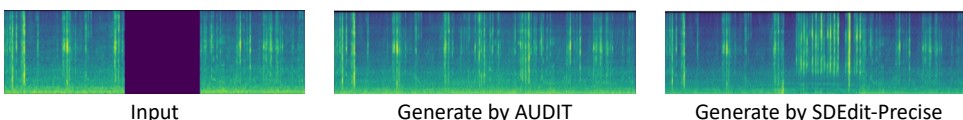

Figure 7: A case for the inpainting task. the caption of the input audio is *"A person is typing computer"*.

## H  Limitations and Broader Impacts

Our work still has some limitations. For example, the sampling efficiency is low since our model is a diffusion-based model. In the future, we will try to improve the generation efficiency of our model using efficient strategies like consistency models [48]. In addition, we will also explore the use of more data and more diverse editing instructions to achieve more kinds of editing tasks. AUDIT can edit existing audio based on natural language instructions, which may be used inappropriately, such as synthesizing fake audio for fraud. Therefore, we urge everyone not to abuse this technology and develop synthetic audio detection tools.

