# OpenReview forum: "AUDIT: Audio Editing by Following Instructions with Latent Diffusion Models"
_NeurIPS.cc/2023/Conference — NeurIPS 2023 poster_

### Official Review · Reviewer_ujZe · 2023-06-21

**Soundness:** 3 good
**Presentation:** 3 good
**Contribution:** 3 good
**Rating:** 8
**Confidence:** 4

**Summary:**

This paper proposes a zero-shot generative editing model for audio signals. The text-based user edits include adding, dropping, replacing, inpainting, and perceptually upsampling audio signals. The audio is not limited to any particular audio domain (e.g., speech or music). Comparisons to recent text-to-audio and zero-shot editing models demonstrate a clear improvement in the state-of-the-art.

**Strengths:**

This is a notable advance in text-to-audio generation as well as audio editing. This paper outlines a framework for general purpose text-based neural audio editing that I could foresee becoming a blueprint for how this type of editing is performed. I imagine many possible subsequent works focusing on quality improvements and further editing capabilities—which there are many.

One of the challenges of producing such an editing system is that each editing capability requires its own evaluation. That is well handled here.

**Weaknesses:**

This is not “the first audio editing model based on human instructions” (line 43-44) as any editing model is technically following the user’s instructions. Consider “the first model to perform semantic audio editing from text-based prompts” or something similar. Also, you could have a machine generate the instructions, so they don’t have to be human users.

The evaluation of super-resolution would benefit from a comparison with a recent task-specific model (e.g., [1, 2]). I’m not yet convinced that performing super-resolution via text prompt would be preferable over having a dedicated super-resolution model.

It is not clear what each of the baseline models is in Table 7.

[1] Su, Jiaqi, Zeyu Jin, and Adam Finkelstein. "HiFi-GAN-2: Studio-quality speech enhancement via generative adversarial networks conditioned on acoustic features." 2021 IEEE Workshop on Applications of Signal Processing to Audio and Acoustics (WASPAA). IEEE, 2021.

[2] Andreev, Pavel, et al. "Hifi++: a unified framework for bandwidth extension and speech enhancement." ICASSP 2023-2023 IEEE International Conference on Acoustics, Speech and Signal Processing (ICASSP). IEEE, 2023.

**Questions:**

Why regenerate the context conditioning during inpainting? Don’t you already have access to the ground truth?

It seems speech is excluded throughout the paper—including in the audio samples. Why exclude speech?

Have you evaluated performance formally or informally on out-of-domain data? What sort of results do you get? How fragile is the model outside the training distribution?

**Limitations:**

Listening to the audio samples, the pitch reconstruction seems noticeably off—enough that the current system would not yet be amenable to fine-grained music editing. As well, the reconstruction of high frequencies could be improved and the context conditioning degrades noticeably in inpainting. The paper could benefit from a reconstruction task (i.e., inpaint with no mask) to better understand why context conditioning is degrading when the ground truth is provided to the model. Otherwise, it's not clear whether quality/accuracy errors can be attributed to the audio generator or the editing process.

---

> ### Author Rebuttal · Authors · 2023-08-09
>
> Thanks for your careful and valuable comments! We will explain your concerns point by point.
>
> > **1. This is not “the first audio editing model based on human instructions” (line 43-44) as any editing model is technically following the user’s instructions. Consider “the first model to perform semantic audio editing from text-based prompts” or something similar. Also, you could have a machine generate the instructions, so they don’t have to be human users.**
>
> Thank you very much for your valuable suggestions! We will revise the wording in the revised version of the paper to make it more accurate.
>
> > **2. The evaluation of super-resolution would benefit from a comparison with a recent task-specific model. I’m not yet convinced that performing super-resolution via text prompt would be preferable over having a dedicated super-resolution model.**
>
> Thanks for your valuable suggestion! Since our approach is based on a diffusion model, our primary comparisons are conducted with zero-shot editing methods that also rely on diffusion models. We have incorporated comparative experiments with task-specific models, focusing on the tasks of super-resolution and inpainting. Please check our general rebuttal or our rebuttal for Reviewer XrSg (Question 1).
>
> > **3. It is not clear what each of the baseline models is in Table 7.**
>
> We have opted to conduct a subjective evaluation on the best-performing baseline methods from Table 1 to Table 5. In the revised version of the paper, we will amend the description in Table 7 to specify the names of the baseline methods that were selected for the subjective evaluation.
>
> > **4. Why regenerate the context conditioning during inpainting? Don’t you already have access to the ground truth?**
>
> We regenerate the context conditioning during inpainting to better assess our model's ability to retain the unedited portions. In fact, for improved results, it is possible to replace non-masked areas with ground truth information (this approach is adopted by our baseline methods, SDEdit-Precise and SDEdit-Rough). For the sake of maintaining overall framework consistency, we omitted this step in the final system.
>
> > **5. It seems speech is excluded throughout the paper—including in the audio samples. Why exclude speech?**
>
> It's a truly thought-provoking question! In reality, our training data includes audio data containing human speech. Our showcase samples also encompass instances like "Drop the sound of a woman talking." However, due to the intricate nature of human speech and the absence of finer supervisory signals (such as phonemes), generating or editing high-quality speech data can be challenging. For instance, even though my training data comprises audio segments of people speaking, when attempting to generate audio of a person speaking, like "A man is speaking and a dog is barking," we can produce speech-like sounds but are unable to discern the specific speech content, making it appear as if it's gibberish.
>
> Our paper primarily concentrates on the broader aspects of general audio generation and editing. In the future, we aspire to seamlessly integrate speech as well, yet I believe that achieving this goal will undoubtedly require supplying more granular supervisory signals.
>
> > **6. Have you evaluated performance formally or informally on out-of-domain data? What sort of results do you get? How fragile is the model outside the training distribution?**
>
> We tested our model on some real data and discovered that as long as the semantic information of the audio events requiring editing (e.g., labels) is present in our training data, our model performs well in executing editing tasks. For instance, it excels at super-resolution on real speech data or inpainting on music data. In the case of adding, dropping, and replacement tasks, our model demonstrates robust recognition capabilities for most audio events and can successfully carry out these edits on real-world data.
>
> We also discussed the model's generalization at the task level. For example, generalization from single tasks to composite tasks. You can check our response to reviewer LhWX's question 2 for more details.
>
> > **7. Limitations.**
>
> We encountered a similar issue where the audio quality slightly degraded after editing. This problem primarily stems from the vocoder and VAE reconstruction. We computed objective metrics for audio generated using ground truth mel spectrograms plus vocoder, as well as VAE reconstruction plus vocoder. Of course, we believe that introducing a complete reconstruction task (directly copying the input) in our training regimen can further aid the model in retaining unedited portions. This suggestion is highly valuable!
>
> | Setting | FD($\downarrow$) | IS($\uparrow$) |KL($\downarrow$)
> | :---: | :---: | :---: | :---: |
> | GT mel + vocoder | 7.11 | 12.02 | 0.18 |
> | VAE reconstruction + vocoder | 9.89 | 14.37 | 0.35 |
> | | | | |

---

> > ### Comment · Reviewer_ujZe · 2023-08-12
> >
> > The authors have made thoughtful consideration towards addressing the weaknesses and limitations mentioned. The evaluation in particular has improved significantly. I think the discussion that the model produces babble when editing speech should make its way into the text. It's a sensible result and won't lead readers to believe you have solved a harder problem than you currently have.

---

> > > ### Author Response · Authors · 2023-08-13
> > >
> > > Thank you for your reply! We will address the issue of speech editing explicitly in the revised version of the paper. Our future research goal is to achieve finer-grained and controllable editing, such as modifying the content of human speech and the style of music.

---

### Official Review · Reviewer_ptM3 · 2023-07-06

**Soundness:** 3 good
**Presentation:** 2 fair
**Contribution:** 2 fair
**Rating:** 4
**Confidence:** 4

**Summary:**

This work proposes a new editing technique to audio data.

**Strengths:**

The authors introduce a new task and a base method to tackling it.

**Weaknesses:**

**Conditioning:** The authors concatenate the input audio to the sampled noise, why did you pick this option on top of sampling a Gaussian around the input sample of giving it as part of the cross attention (which will probably allow for modifications in the generation length)? Can the authors elaborate on this point?

**FAD** The authors do not show the FAD of audiogen and make-an-audio. Why? I noted that the classifier is different (PANN vs. VGGISH), but it is hard to understand what are the differences in performance.

**Comparison to prior work:** I am having hard time understanding what is the cause for the model's performance - is it the data or the modeling. All the baselines present results when trained on different (mostly open) datasets; To do an appropriate comparison, it is expected to at least train one-two of those models on the same data the authors used.

**Subjective study** I find the experimental study not reproducible (even after reading Appendix D). Did the authors also check metrics compared to ground truth data? At least for dropping and super resolution, you have a hard gt data.

**Error bars** Some of the metric differences are small - can the authors report error bars?

Overall: I think that the task is interesting. However, since it is a new task and the paper doesn't introduce a new modeling approach, I would expect extensive experiments, which I was unable to find.

**Questions:**

**Technical questions:**
- Which diffusion process was used?
- How do you pass the condition to the model?

I couldn't find answers to those questions explicitly in the paper.

---

> ### Author Rebuttal · Authors · 2023-08-09
>
> Thanks for your careful and valuable comments! We will explain your concerns point by point.
>
> > **1. conditioning.**
>
> Thank you found the alternative conditioning approaches helpful. Using Gaussian sampling around the input sample as the starting point for diffusion denoising (replacing the conventional method of directly sampling noise from a standard normal distribution) is indeed a viable approach, as demonstrated in works like GradTTS. However, it's worth noting that this approach might not provide a completely origin input audio due to the presence of variance, So, this approach might not be the optimal choice.
>
> Using the input audio as cross-attention input would indeed introduce additional computational complexity. On the other hand, directly concatenating the latent representation of the input audio along the channel dimension is a simpler and more straightforward approach. This design allows the model to potentially "copy" the content of the input audio in regions where no editing is required. This simplicity can also aid in the preservation of unedited areas.
>
> This approach is also present in some other works, such as the image editing model Palette [1]. We believe that this method represents one of the most straightforward and simplistic ways to ensure that the model can comprehensively "see" the audio that needs to be edited during both training and inference.
>
> > **2. FAD.**
>
> Our subjective evaluation criteria follow AudioLDM. AudioLDM points out that the difference between the FD and FAD metrics lies in the use of a more advanced audio classification model, PANNs, for FD. Therefore, FD could potentially be a better evaluation metric. As a result, we have chosen FD as the main evaluation metric in our paper. We will supplement the results of AUDIT on FAD in the revised version of the paper. You can refer to the table below question 3 for more details!
>
> > **3. Comparison to prior work.**
>
> Compared to previous work, our text-to-audio model has a larger parameter count. Our diffusion U-Net has 859M parameters (while AudioLDM-L has 739M). Additionally, we observed a significant impact of the vocoder on objective metrics. A vocoder trained with larger datasets and longer training times can improve objective metrics.
>
> In essence, our paper primarily focuses on audio editing by following instructions. We trained the text-to-audio diffusion model to provide a strong baseline (the SDEdit method based on the text-to-audio diffusion model) for our editing model. Therefore, we aimed to achieve high performance with our generation model. Indeed, our text-to-audio model has reached state-of-the-art levels, ensuring its effectiveness as a baseline for the editing model.
>
> To conduct a fair comparison for the text-to-audio generation model, we also included results from models trained on AudioCaps and AudioSet datasets. However, it's important to note that previous work using the AudioSet dataset also involved using only a subset of AudioSet (e.g., excluding speech, music, etc.). Due to the lack of specific data segmentation details, achieving complete data parity remains challenging. You can find more details in the table below.
>
> | Model | Dataset | FD($\downarrow$) | IS($\uparrow$) |KL($\downarrow$) | FAD($\downarrow$) |
> | :---: | :---: | :---: | :---: | :---: | :---: |
> | DiffSound | AudioSet+AudioCaps | 47.68 | 4.01 | 2.52 | 7.75|
> | AudioGen |  AudioSet+AudioCaps | - | - | 2.09| 3.13|
> | Make-an-Audio | AudioSet+AudioCaps+13 others | - | - | 2.79| 4.61|
> | AudioLDM-L |  AudioCaps | 27.12| 7.51| 1.86| 2.08|
> | AudioLDM-L-Full | AudioSet+AudioCaps+2 others | 23.31| 8.13| 1.59| **1.96**|
> | AUDIT | AudioSet+AudioCaps+2 others| **20.19**| **9.23**| **1.32**| 2.01|
> | AUDIT | AudioSet + AudioCaps| 21.24| 8.47| 1.39| 1.98|
> | | | | | | |
>
> > **4. Subjective study**
>
> To evaluate audio quality, we explicitly instruct the raters to "focus on examining the audio quality", to evaluate text-audio alignment, raters are shown an audio and a text and asked "Does the natural language description align with the audio". For the super-resolution and inpainting tasks, we also evaluated the results using ground truth mel spectrograms (Inpainting: Quality 89.74, Relevance 91.92; Super-resolution: Quality 90.31, Relevance 92.03). We will add the result in the revised version paper.
>
> > **5. Error bars**
>
> When comparing to prior work, we've seen substantial gains in objective metrics, You can check the table below question 3 to get more details.
>
> > **6. Which diffusion process was used?**
>
> We employ a DDPM process for the forward diffusion process, where the total number of forward diffusion steps is set to 1000 (with $\beta_0=0.0001, \beta_{1000}=0.02$, and a linear beta schedule). In the reverse process, we utilize DDIM for sampling. The default number of sampling steps for DDIM is set to 100.
>
> > **7. How do you pass the condition to the model?**
>
> For the text-to-audio generation model, where the condition is text, we begin by encoding the textual content into a text embedding using a T5 encoder model. This text embedding is then incorporated into the model using cross-attention, the text embedding providing the keys (K) and values (V) for this cross-attention mechanism.
>
> Regarding the audio editing model, where the conditions are text and input audio, the text condition is introduced in the same manner as in the text-to-audio model. As for the input audio, we directly concatenate the latent representation of the input audio with the model's input, which consists of the latent representation of the target audio after introducing noise. This concatenation is performed along the channel dimension, ensuring that the model has access to information from the input audio during both training and inference stages. This design allows the model to "see" the input audio and effectively utilize it for the editing process.
>
> [1] Saharia C, Chan W, Chang H, et al. Palette: Image-to-image diffusion models.

---

> > ### Author Response · Authors · 2023-08-14
> >
> > Hi, have the issues been resolved, and are there any further questions? We are available to address them.

---

> > > ### Comment · Reviewer_ptM3 · 2023-08-19
> > >
> > > I thank the authors for their elaborated response.
> > >
> > > Re. Conditioning. Let me clarify my question: (i) DDPM process relies on the promise that its input is sampled from a Gaussian distribution, and you decided to ignore this assumption. I understand that it works, my question is - is it optimal? (ii) Potential copy - did you check this as well? For example, did you see that the model even consider the noise? For example, through looking at the gradients of the output w.r.t the noise? Is it possible that your model completely ignores the noise? Following this, did you evaluate if the model does copy the input? i.e., in remove - does it keep the original input but the channel you remove?
> > >
> > > Re. FAD and comparison. If I understand your table correctly, I am unsure that your comparison is fair w.r.t AudioGen as you report their small model and your model is much closer to their large one, that is better in terms of FAD.

---

> > > > ### Author Response · Authors · 2023-08-20
> > > >
> > > > Thank you for clarifying your question! Since our work is based on the diffusion model, we assume that the starting point of the reverse diffusion process is standard Gaussian noise. Since we hope that the parts that do not need to be edited will not be edited as much as possible, we allow the model to "see" input audio all the time, which means we encourage the model to directly copy the parts that do not need to be edited. We also found that the model can achieve this, for example, the low frequency remains unchanged in the super-resolution task, and the unmask part remains unchanged in the inpainting task. Regarding the comparison of FAD, we will add the comparison with audiogen-large (our model FAD is slightly lower than audiogen-large, but KL performs better).

---

### Official Review · Reviewer_LhWX · 2023-07-06

**Soundness:** 4 excellent
**Presentation:** 4 excellent
**Contribution:** 4 excellent
**Rating:** 5
**Confidence:** 5

**Summary:**

This paper proposes AUDIT, an instruction-guided audio editing model based on latent diffusion models. AUDIT is trained on triplet data consisting of an instruction, an input audio, and an output audio. The instruction is used to guide the diffusion model to only modify the audio segments that need to be edited. AUDIT achieves state-of-the-art results in both objective and subjective metrics for several audio editing tasks, such as adding, dropping, replacement, inpainting, and super-resolution.

**Strengths:**

AUDIT is a novel audio editing model that can be trained using human text instructions. AUDIT is trained on a dataset of triplet data, which consists of an audio clip, a text instruction, and the edited audio clip. AUDIT can maximize the preservation of audio segments that do not need to be edited, and it only requires simple instructions as text guidance. AUDIT has achieved state-of-the-art results in both objective and subjective metrics for several audio editing tasks.



**Weaknesses:**

.

**Questions:**

1. Please add ablation analysis for each component.
2. Can you train it on one task and try to zero shot in other task?
3. The model compare to SDEdit. please add more comparison to other method (classic / deep learning based).

**Limitations:**

.

---

> ### Author Rebuttal · Authors · 2023-08-09
>
> Thanks for your careful and valuable comments! We will explain your concerns point by point.
>
> > **1. Please add ablation analysis for each component.**
>
> We have supplemented the impact of the text encoder on the performance of the text-to-audio generation model.
>
> | Text Encoder | FD($\downarrow$) | IS($\uparrow$) |KL($\downarrow$)
> | :---: | :---: | :---: | :---: |
> | T5-base | **20.19** | **9.23** | **1.32** |
> | CLAP | 21.62| 8.25| 1.51|
> | BERT-base | 20.97| 9.12| 1.37|
> | | | | |
>
> We have observed that language models pre-trained solely on textual data as text encoders achieve better performance compared to the CLAP model. Although the CLAP model is trained on text-audio pairs, its representation of audio collapses into a single embedding, providing a relatively coarse audio semantic information. As a result, it might struggle to effectively represent complex audio containing multiple events, including the relationships between different audio events, such as sequential order. For instance, consider the example: "A man is speaking, after that, a train is passing by."
>
> We also conducted a comparison of our vocoder's impact on model objective metrics, contrasting it with AudioLDM. Our vocoder exhibited superior performance, which contributed to the enhancement of our model's objective metrics in the realm of text-to-audio generation.
>
> | Vocoder | FD($\downarrow$) | IS($\uparrow$) |KL($\downarrow$)
> | :---: | :---: | :---: | :---: |
> | AudioLDM | 8.76 | 10.71 | 0.23 |
> | Ours | **7.11** | **12.02** | **0.18** |
> | | | | |
>
> For our audio editing model, we compared the differences between instruction prompts generated using ChatGPT and instructions generated directly through rules. We found that instruction prompts generated by ChatGPT tend to yield better results compared to instructions generated solely by rules. This is primarily because instruction data generated by ChatGPT exhibits a richer distribution, while instructions directly generated by rules tend to have a more limited distribution.
>
> > **2. Can you train it on one task and try to zero shot in other task?**
>
> A crucial issue! In our work, we primarily consider five different types of editing tasks, while in reality, there are many specific types of editing tasks. Can we build a general audio editing model that can handle almost all editing tasks, similar to the current large language models that can process a variety of tasks based on provided prompts? This is an important direction for our future work. For the current version of AUDIT, our model demonstrates a certain level of generalization at the task level:
>
> - Generalization from Single Tasks to Composite Tasks. Our training data consists of five distinct editing tasks, with each data point focusing on one of the five tasks. Our model demonstrates a certain degree of generalization to composite tasks. For instance, when presented with a segment of low-resolution audio along with a portion masked, we can simultaneously apply super-resolution and completion to the audio.
>
> - Generalization of Editing Audio Event Count. A significant portion of our dataset involves editing a single audio event, such as "add a dog barking sound to the audio." Our model also exhibits generalization to editing multiple audio events simultaneously. For example, given the instruction "add a dog barking sound and a baby crying," our model can simultaneously incorporate both a dog barking sound and a baby crying sound.
>
> > **3. Please add more comparison to other method (classic / deep learning based).**
>
> we have add comparison to task-specific methods for super-resolution and inpainting.
>
> Super-resolution:
>
> For the super-resolution task, we conducted a comparison with the NU-Wave2 [1] model, which is a task-specific model designed for super-resolution. We utilized the provided open-source checkpoint for NU-Wave2; however, note that NU-Wave2 was originally trained on speech data. To adapt it to our specific task, we trained it on our own dataset.
>
> | Mehtod | LSD($\downarrow$) | KL($\downarrow$) |FD($\downarrow$)
> | :---: | :---: | :---: | :---: |
> |SDEdit (best)|1.75|1.17|27.81|
> | AUDIT | 1.48| **0.73**| **18.14**|
> | NU-Wave2 (origin) | 1.66| 0.89| 22.61|
> | NU-Wave2 (trained with our data)| **1.42**| 0.78| 19.57|
> | | | | |
>
> The results indicate that our AUDIT model outperforms the NU-Wave2 model trained on speech and is comparable to or even better than the NU-Wave2 model trained on our dataset.
>
> Inpainting:
>
> For the audio inpainting task, we conducted comparative experiments with GACELA [2] and DAS [3]. GACELA is a generative adversarial network for restoring missing musical audio data. The publicly available checkpoint was trained solely on music data; hence, we trained the model using our own dataset. DAS is another GAN-based audio editing model, we also trained it on our dataset.
>
> | Mehtod | LSD($\downarrow$) | KL($\downarrow$) |FD($\downarrow$)
> | :---: | :---: | :---: | :---: |
> |SDEdit (best)|1.54|0.94|21.07|
> | AUDIT | **1.32**| **0.75**| **18.17**|
> | GACELA (trained with our data) | 1.41| 0.78| 20.49|
> | DAS (trained with our data) | 1.57 | 0.89| 21.97|
> | | | | |
>
> We discovered that AUDIT's performance on inpainting tasks can surpass that of task-specific baseline methods. One possible explanation for this is that during training, GACELA and DAS do not have access to any textual information, whereas our model requires explicit instructions to edit the audio. As a result, our model is better able to learn the semantic information of the audio, which likely contributes to its improved performance in inpainting tasks.
>
> We will add more experiments in the revised version!
>
> [1] Han et al., NU-Wave 2: A General Neural Audio Upsampling Model for Various Sampling Rates
>
> [2] Marafioti A, Majdak P, Holighaus N, et al. A generative adversarial context encoder for long audio inpainting[J]. 2020.
>
> [3] Ebner P P, Eltelt A. Audio inpainting with generative adversarial network[J]. arXiv preprint arXiv:2003.07704, 2020.

---

> > ### Author Response · Authors · 2023-08-14
> >
> > Hi, have the issues been resolved, and are there any further questions? We are available to address them.

---

### Official Review · Reviewer_XrSg · 2023-07-07

**Soundness:** 3 good
**Presentation:** 3 good
**Contribution:** 2 fair
**Rating:** 5
**Confidence:** 4

**Summary:**

This paper proposes AUDIT, an instruction-guided audio editing model based on latent diffusion models. The model is trained with triplet training data (instruction, input audio, output audio) for
different audio editing tasks and is based on a diffusion model using instruction and input
(to be edited) audio as conditions and generating output (edited) audio. The proposed model outperforms several baselines in the editing tasks

**Strengths:**

The AUDIT model is able to perform audio editing conditioned on text instructions, which have rarely been studied by prior work on audio generation. The data construction strategy of using paired text-audio data and refining text instructions based on templates and LLM is also useful for making instruction datasets. In terms of performance, it is better than several baselines (SDEdit and its variants).

**Weaknesses:**

The main weakness lies in the evaluation. The model is mostly compared to SDEdit and its variants, which is zero-shot audio editing model. It is unclear how it compares to task-specific models such as NU-Wave2 [1] for audio super-resolution. Some of the editing tasks considered in the paper (e.g., Adding) can also be addressed via text-to-audio generation, while proposed method has not been compared to the methods of this category. There also lacks analysis on the gain of the proposed method over SDEdit and its variants. For example, how robust the model is to the category of the dropped sound in Dropping task? As both the training and evaluation data are synthesized, it is unclear how the model will perform on real data.

[1]. Han et al., NU-Wave 2: A General Neural Audio Upsampling Model for Various Sampling Rates

**Questions:**

N/A

**Limitations:**

Yes

---

> ### Author Rebuttal · Authors · 2023-08-05
>
> Thanks for your careful and valuable comments! We will explain your concerns point by point.
>
> >**1. It is unclear how it compares to task-specific models such as NU-Wave2 [1] for audio super-resolution.**
>
> Thanks for your valuable suggestion! Since our approach is based on a diffusion model, our primary comparisons are conducted with zero-shot editing methods that also rely on diffusion models. We have incorporated comparative experiments with task-specific models, focusing on the tasks of super-resolution and inpainting.
>
> Super-resolution:
>
> For the super-resolution task, we conducted a comparison with the NU-Wave2 [1] model, which is a task-specific model designed for super-resolution. We utilized the provided open-source checkpoint for NU-Wave2; however, note that NU-Wave2 was originally trained on speech data. To adapt it to our specific task, we trained it on our own dataset.
>
> | Mehtod | LSD($\downarrow$) | KL($\downarrow$) |FD($\downarrow$)
> | :---: | :---: | :---: | :---: |
> |SDEdit (best)|1.75|1.17|27.81|
> | AUDIT | 1.48| **0.73**| **18.14**|
> | NU-Wave2 (origin) | 1.66| 0.89| 22.61|
> | NU-Wave2 (trained with our data)| **1.42**| 0.78| 19.57|
> | | | | |
>
> The results indicate that our AUDIT model outperforms the NU-Wave2 model trained on speech and is comparable to or even better than the NU-Wave2 model trained on our dataset.
>
> Inpainting:
>
> For the audio inpainting task, we conducted comparative experiments with GACELA [2] and DAS [3]. GACELA is a generative adversarial network for restoring missing musical audio data. The publicly available checkpoint was trained solely on music data; hence, we trained the model using our own dataset. DAS is another GAN-based audio editing model, we also trained it on our dataset.
>
> | Mehtod | LSD($\downarrow$) | KL($\downarrow$) |FD($\downarrow$)
> | :---: | :---: | :---: | :---: |
> |SDEdit (best)|1.54|0.94|21.07|
> | AUDIT | **1.32**| **0.75**| **18.17**|
> | GACELA (trained with our data) | 1.41| 0.78| 20.49|
> | DAS (trained with our data) | 1.57 | 0.89| 21.97|
> | | | | |
>
> We discovered that AUDIT's performance on inpainting tasks can surpass that of task-specific baseline methods. One possible explanation for this is that during training, GACELA and DAS do not have access to any textual information, whereas our model requires explicit instructions to edit the audio. As a result, our model is better able to learn the semantic information of the audio, which likely contributes to its improved performance in inpainting tasks.
>
> > **2. Some of the editing tasks considered in the paper (e.g., Adding) can also be addressed via text-to-audio generation.**
>
> Indeed, tasks such as adding, dropping, and replacement editing can be achieved through text-to-audio generation. However, directly using a text-to-audio generation model might struggle to retain the parts that don't require editing. For example, if we want to add "adding a train is passing by" to the audio of "a man is speaking," using a text-to-generation model would require inputting "a man is speaking while a train is passing by." But in the resulting new audio generated by the model, the "a man is speaking" part would changed.
>
> In fact, we have actually compared it with the text-to-generation model. We employ SDEdit as the baseline for our editing tasks. We add noise and then use a pre-trained text-to-audio diffusion model to denoise input audio with the editing target description. When SDEdit uses a large number of noising steps, denoted as $N$ (such as when $N$ equals the total steps $T$ in the forward diffusion process used to train the text-to-audio generation model), it can be viewed as directly generating new audio from the introduced noise. In our paper, we present the results of the baseline method (SDEdit) at different values of $N$, we observed that the editing performance of these variations of SDEdit consistently outperforms the direct generation approach using our trained text-to-audio model (or equivalently, setting $N$ equal to $T$ within SDEdit).
>
> > **3. There also lacks analysis on the gain of the proposed method over SDEdit and its variants.**
>
> In Section 3.3 of the paper, we analyze the advantages of the AUDIT model. In contrast to the zero-shot SDEdit based on a text-to-audio generative diffusion model, our method directly use the input audio as a condition for supervised training of our diffusion model, allowing the model to automatically learn to preserve the parts that do not need to be edited before and after editing.  In contrast, SDEdit requires editing to commence from the noised input audio based on new textual descriptions, lacking visibility into the origin audio. Furthermore, SDEdit can't engage in editing directly based on commands, unlike AUDIT (which can operate based on concise instructions), it needs a complete description of the target. In the revised version, we will provide a general summary encompassing the aforementioned aspects.
>
> > **4. How robust the model is on real data.**
>
> We tested our model on some real data and discovered that as long as the semantic information of the audio events requiring editing (e.g., labels) is present in our training data, our model performs well in executing editing tasks. For instance, it excels at super-resolution on real speech data or inpainting on music data. In the case of adding, dropping, and replacement tasks, our model demonstrates robust recognition capabilities for most audio events and can successfully carry out these edits on real-world data.
>
> We also discussed the model's generalization at the task level. You can check our response to reviewer LhWX's question 2 for more details.
>
> [1] Han et al., NU-Wave 2: A General Neural Audio Upsampling Model for Various Sampling Rates
>
> [2] Marafioti A, Majdak P, Holighaus N, et al. A generative adversarial context encoder for long audio inpainting[J]. 2020.
>
> [3] Ebner P P, Eltelt A. Audio inpainting with generative adversarial network[J]. arXiv preprint arXiv:2003.07704, 2020.

---

> > ### Author Response · Authors · 2023-08-14
> >
> > Hi, have the issues been resolved, and are there any further questions? We are available to address them.

---

> > > ### Comment · Reviewer_XrSg · 2023-08-21
> > >
> > > I thank the authors for their response. Regarding the model performance on real data, could you elaborate on adding/dropping/replacement? What data did you use and how did you evaluate the model performance?

---

> > > > ### Author Response · Authors · 2023-08-21
> > > >
> > > > We mainly use audiocaps containing two or more audio data for testing, and the main evaluation standard is human subjective evaluation. We found that our model can handle real data relatively well.

---

### Author Rebuttal · Authors · 2023-08-09

Thank you very much for the valuable suggestions and insightful questions you've provided! We have addressed each reviewer's concerns point by point.

Furthermore, we would like to reiterate some common points of discussion (of course, we have also responded individually to each reviewer's specific questions).

**Regarding the comparison with task-specific models**, we have expanded our experiments to include audio super-resolution and audio inpainting tasks.

For the super-resolution task, we conducted a comparison with the NU-Wave2 [1] model, which is a task-specific model designed for super-resolution. We utilized the provided open-source checkpoint for NU-Wave2; however, it's important to note that NU-Wave2 was originally trained on speech data. To adapt it to our specific task, we performed additional training on our own dataset.

| Mehtod | LSD($\downarrow$) | KL($\downarrow$) |FD($\downarrow$)
| :---: | :---: | :---: | :---: |
|SDEdit (best)|1.75|1.17|27.81|
| AUDIT | 1.48| **0.73**| **18.14**|
| NU-Wave2 (origin) | 1.66| 0.89| 22.61|
| NU-Wave2 (trained with our data)| **1.42**| 0.78| 19.57|
| | | | |

The results indicate that our AUDIT model outperforms the NU-Wave2 model trained on speech and is comparable to or even better than the NU-Wave2 model trained on our dataset.

For the audio inpainting task, we conducted comparative experiments with GACELA [2] and DAS [3]. GACELA is a generative adversarial network for restoring missing musical audio data. The publicly available checkpoint was trained solely on music data; hence, we trained the model using our own dataset. DAS is another GAN-based audio editing model, we also trained it on our dataset.

| Mehtod | LSD($\downarrow$) | KL($\downarrow$) |FD($\downarrow$)
| :---: | :---: | :---: | :---: |
|SDEdit (best)|1.54|0.94|21.07|
| AUDIT | **1.32**| **0.75**| **18.17**|
| GACELA (trained with our data) | 1.41| 0.78| 20.49|
| DAS (trained with our data) | 1.57 | 0.89| 21.97|
| | | | |


We discovered that AUDIT's performance on inpainting tasks can surpass that of task-specific baseline methods. One possible explanation for this is that during training, GACELA and DAS do not have access to any textual information, whereas our model requires explicit instructions to edit the audio. As a result, our model is better able to learn the semantic information of the audio, which likely contributes to its improved performance in inpainting tasks.

**Regarding the generalization of our model**, we have primarily discussed two key aspects. Firstly, we have examined its generalization to real data or out-of-domain scenarios. Secondly, we have explored its generalization capabilities in various editing tasks.

We tested our model on some real data and discovered that as long as the semantic information of the audio events requiring editing (e.g., labels) is present in our training data, our model performs well in executing editing tasks. For instance, it excels at super-resolution on real speech data or inpainting on music data. In the case of adding, dropping, and replacement tasks, our model demonstrates robust recognition capabilities for most audio events and can successfully carry out these edits on real-world data.

In our work, we primarily consider five different types of editing tasks, while in reality, there are many specific types of editing tasks. Can we build a general audio editing model that can handle almost all editing tasks, similar to the current large language models that can process a variety of tasks based on provided prompts? This is an important direction for our future work. For the current version of AUDIT, our model demonstrates a certain level of generalization at the task level:

- Generalization from Single Tasks to Composite Tasks. Our training data consists of five distinct editing tasks, with each data point focusing on one of the five tasks. Our model demonstrates a certain degree of generalization to composite tasks. For instance, when presented with a segment of low-resolution audio along with a portion masked, we can simultaneously apply super-resolution and completion to the audio.

- Generalization of the Number of Audio Events to be Edited. A significant portion of our dataset involves editing a single audio event, such as "add a dog barking sound to the audio." Our model also exhibits generalization to editing multiple audio events simultaneously. For example, given the instruction "add a dog barking sound and a baby crying," our model can simultaneously incorporate both a dog barking sound and a baby crying sound.

The primary innovations of our work lie in introducing a novel task format, performing semantic audio editing from text-based prompts, and establishing a pipeline that leverages large language models to construct training datasets. In the future, our objective is to unify a broader spectrum of audio generation and editing tasks under a prompt-guided framework. We hope that our efforts can serve as an initial exploration for subsequent research in the community.

If you have any further questions or concerns, please do not hesitate to let us know. We are more than happy to engage in further discussions and exchanges with all of you!

---

### Decision · Program_Chairs · 2023-09-21

**Decision:**

Accept (poster)

**Comment:**

All reviewers agree that the results presented by the authors are impressive and the proposed method is interesting. The authors engaged with reviewers during the rebuttal phase and provided additional results and clarifications which makes this submission stronger.
The main concern raised by the reviewers is model evaluation using real-world data. I do encourage the authors to include the additional details and results provided during the rebuttal period in their final version, including results on real-world data. Due to all of the above, I recommend acceptance.